# Neural Attention Field: Emerging Point Relevance in 3D Scenes for One-Shot Dexterous Grasping

**Qianxu Wang**
Peking University

**Congyue Deng**
Stanford University

**Tyler Lum**
Stanford University

**Yuanpei Chen**
Peking University

**Yaodong Yang**
Peking University

**Jeannette Bohg**
Stanford University

**Yixin Zhu**
Peking University

**Leonidas Guibas**
Stanford University

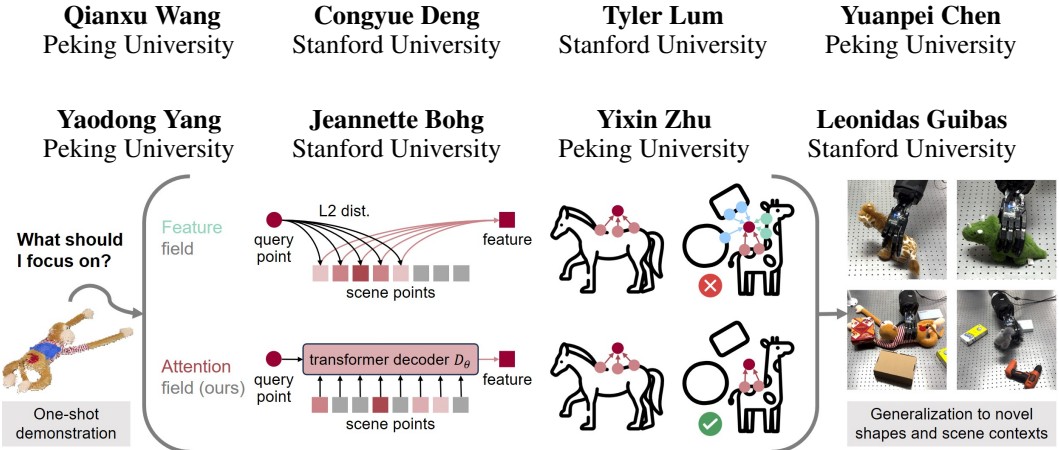

Figure 1: Given a one-shot demonstration of a dexterous grasp, we want to generalize to novel scene variations with relevant semantics. To better model the complex semantic feature distributions in hand-object interactions, we propose the **neural attention field**, which represents a semantic-aware dense feature field by modeling inter-point relevance instead of individual point features. It encourages the end-effector to focus on scene regions with higher task relevance instead of spatial proximity, resulting in robust and semantic-aware transfer of dexterous grasps across scenes.

**Abstract:** One-shot transfer of dexterous grasps to novel scenes with object and context variations has been a challenging problem. While distilled feature fields from large vision models have enabled semantic correspondences across 3D scenes, their features are point-based and restricted to object surfaces, limiting their capability of modeling complex semantic feature distributions for hand-object interactions. In this work, we propose the *neural attention field* for representing semantic-aware dense feature fields in the 3D space by modeling inter-point relevance instead of individual point features. Core to it is a transformer decoder that computes the cross-attention between any 3D query point with all the scene points, and provides the query point feature with an attention-based aggregation. We further propose a self-supervised framework for training the transformer decoder from only a few 3D pointclouds without hand demonstrations. Post-training, the attention field can be applied to novel scenes for semantics-aware dexterous grasping from one-shot demonstration. Experiments show that our method provides better optimization landscapes by encouraging the end-effector to focus on task-relevant scene regions, resulting in significant improvements in success rates on real robots compared with the feature-field-based methods.

**Keywords:** Dexterous Grasping, One-Shot Manipulation, Distilled Feature Field, Neural Implicit Field, Self-Supervised Learning

## 1 Introduction

Generalization has been a long-studied problem in robot learning from demonstrations, with one-shot generalization being one of the most challenging settings. Great progress has been made with recent advances in large vision models which provide visual features with cross-scene semantic correspondences [1, 2]. Given a pre-trained vision model on 2D images, a common strategy is to

8th Conference on Robot Learning (CoRL 2024), Munich, Germany.

distill its features into 3D and match the end-effector position in the target scene to the demonstration by minimizing their feature differences [3, 4, 5, 6].

However, the distilled feature fields can only model point correspondences on the object surfaces. While this is sufficient for simple manipulations with a parallel gripper whose interactions with objects are primarily based on individual contact points, this is insufficient for more complex end-effectors. For example, the hand-object interactions required for dexterous grasping are no longer point-based or limited to the object surface, but instead spatially distributed or even possibly off the object surface. Prior dexterous grasping method with feature fields [6] computes query features in the free space as a sum of closeby scene-point features weighted by distances (Fig. 1 top). While it achieves reliable results in single-object scenes, it is prone to failure when there are distractors in the vicinity of the target object due to its naive feature aggregation based on spatial proximity.

To design a better feature representation, we start with the intuition that, when humans interact with a scene, an intention is formed even before their hands reach the target object. Driven by this intention, they have their attention focused on certain scene regions with higher task relevance instead of looking into the details of every part of the scene. Following this intuition, we propose the *neural attention field* for representing semantic-aware dense feature fields in the 3D space by modeling inter-point relevance instead of individual point features (Fig. 1). More concretely, we replace the distance-weighted feature interpolation in [6] with a transformer decoder that computes the cross-attention between any 3D query point with all the scene points, and aggregates the scene features to the query point based on the attention rather than spatial proximity (Fig. 1 bottom).

We further propose a self-supervised framework for training the transformer decoder from only a few 3D scenes without hand demonstrations (for example, the 4 pointclouds in Fig. 2 left). Here we take the scene points as queries and learn a self-attention on the scene pointcloud. Each scene point itself carries a raw feature, which is updated through the transformer aggregation within each scene. Our key intuition is that, point correspondences can be computed across scenes via feature similarities, and they should stay identical for both features before and after applying the transformer. This leads to a contrastive feature loss for optimizing the transformer decoder parameters.

Post-training, the transformer decoder can be directly applied to novel scenes for querying the feature of any 3D point in the near-object space. We can then use it to transfer one-shot demonstration of dexterous grasps from a source scene to a target scene following the same strategy as neural feature fields [7, 6], where we sample query points on the hand surface and minimize their feature differences between the two scenes.

Experimentally, we show that our neural attention field induces better optimization landscapes by encouraging the end-effector to focus on task-related scene regions and omitting distractions, enabling robust one-shot dexterous grasping in various challenging scenarios such as grasping objects in complex scene contexts, grasp transfer between objects of varying shapes but similar semantics, or functional grasping of tools. Real-robot results demonstrate our significant improvements in success rates compared with the feature-field-based methods.

To summarize, our key contributions are: (i) We propose the *neural attention field* for representing semantic-aware dense feature fields in the 3D space by modeling inter-point relevance instead of individual point features. (ii) We propose a self-supervised framework for training the attention field with a few scenes. (iii) Our attention field enables semantics-aware dexterous grasping from one-shot demonstration and encourages the end-effector to focus on task-related scene regions. (iv) Real-robot experiments show our strong generalization under different scene variations.

## 2    Related Work

**Neural field for manipulation**    Point correspondences induced by features facilitate the transfer of manipulation policies across diverse objects. In contrast to methods based on key points [8, 9, 10], recent efforts focus on developing dense feature fields around 3D scenes through implicit representations [7, 11, 12, 13, 14, 15, 16, 17, 18, 19]. The integration of large vision models have propelled

research towards leveraging distilled features fields for enabling few-shot or one-shot learning of 6-DOF grasps [3], sequential actions [4], and language-guided manipulations [5]. These efforts predominantly focus on simple manipulations using parallel grippers. Closest to us is [6] which uses distilled feature fields to transfer dexterous grasps across objects. However, as mentioned, it computes query features in the free space as a sum of closeby scene-point features weighted by distances, making it unstable to distractors near the target object.

**Distilled feature field**   With the recent advances of large vision models on 2D images, an effective way of acquiring semantics-aware feature fields for 3D scenes is to first obtain 2D image features and then distill them to 3D. [20, 21, 22] lifts of semantic information from 2D segmentation networks to 3D, enabling 3D segmentations with language embeddings. [23, 24] delve into integrating pixel-aligned image features from models like LSeg or DINO [1] into 3D NeRFs, highlighting their impact on manipulating 3D geometry. Further, [25] and [26] explore distilling non-pixel-aligned image features, such as those from CLIP [27], into 3D scenes without the need for fine-tuning.

**Dexterous grasping**   Dexterous manipulation, central to advanced robotic applications, necessitates nuanced understanding and control, akin to human-like grasping capabilities [28, 29, 30, 31, 32, 33, 34]. Analytical approaches [35, 29, 30] focus on direct modeling of hand and object dynamics. Recent learning-based methods have introduced state [36, 37, 38, 39, 40] and vision-based strategies [41, 42, 43, 44, 45, 46], targeting realistic scene comprehensions. However, most of these methods depend on large demonstration datasets for training. Notably, [47] proposes a grasp synthesis algorithm focusing on geometric and physical constraints for functional grasping that generalizes within similar object shapes using minimal demonstrations,

## 3   Method

### 3.1   Preliminaries

**Demonstration transfer with feature field**   Given a 3D scene pointcloud $\mathbf{X}$, a feature field around the scene is a function $\mathbf{f}(\cdot, \mathbf{X}) : \mathbb{R}^3 \to \mathbb{R}^C$ that maps every query point $\mathbf{q} \in \mathbb{R}^3$ in the 3D space to a $C$-dimension feature $\mathbf{f} \in \mathbb{R}^C$. For an end-effector placed in the feature field with pose parameters $\beta$, one can sample a set of query points $\mathbf{Q}|\beta \in \mathbb{R}^{Q \times 3}$ on the end-effector surface conditioned on its parameters and obtain their features $\mathbf{f}(\mathbf{Q}|\beta, \mathbf{X}) \in \mathbb{R}^{Q \times C}$, which can be viewed as the feature for the entire end-effector *w.r.t.* $\beta$. Given a demonstration with a source pointcloud $\hat{\mathbf{X}}$ and end-effector parameters $\hat{\beta}$, one can transfer it to a target scene $\mathbf{X}$ by minimizing an energy function induced by the two feature fields

$$\arg\min_{\beta} E(\beta) = \arg\min_{\beta} \|\mathbf{f}(\mathbf{Q}|\hat{\beta}, \hat{\mathbf{X}}) - \mathbf{f}(\mathbf{Q}|\beta, \mathbf{X})\|_1. \tag{1}$$

Intuitively, this minimizes the feature differences of the end-effector positions in the scene, indicating that the end-effector should be applied to the locations with similar semantic meanings. More elaborations can be found in [7, 6].

**Distilled feature field**   An effective approach for acquiring a semantic-aware 3D feature field is to first extract semantic features on 2D images with large vision models [1, 2] and then distill them to 3D. A line of works [26, 5, 19] directly reconstruct continuous implicit feature fields alongside NeRF representations. If given a 3D pointcloud and its point correspondences to the 2D image pixels derived from RGBD sensor inputs, one can also directly back project and aggregate the image features on each pixel to the 3D points. The discrete pointcloud features can be further propagated into the surrounding space via spatial interpolation [6]. For a pointcloud $\mathbf{X} = \{\mathbf{x}_i\}$ with per-point features $\mathbf{F} = \{\mathbf{f}_i\}$, the feature $\mathbf{z}$ at a query point $\mathbf{q} \in \mathbb{R}^3$ is interpolated from the pointcloud features with inverse distance weighting:

$$\mathbf{f} = \sum_{i=1}^{N} w_i \mathbf{f}_i, \quad \text{where} \quad w_i = \frac{1/\|\mathbf{q} - \mathbf{x}_i\|^2}{\sum_{j=1}^{N} 1/\|\mathbf{q} - \mathbf{x}_j\|^2}. \tag{2}$$

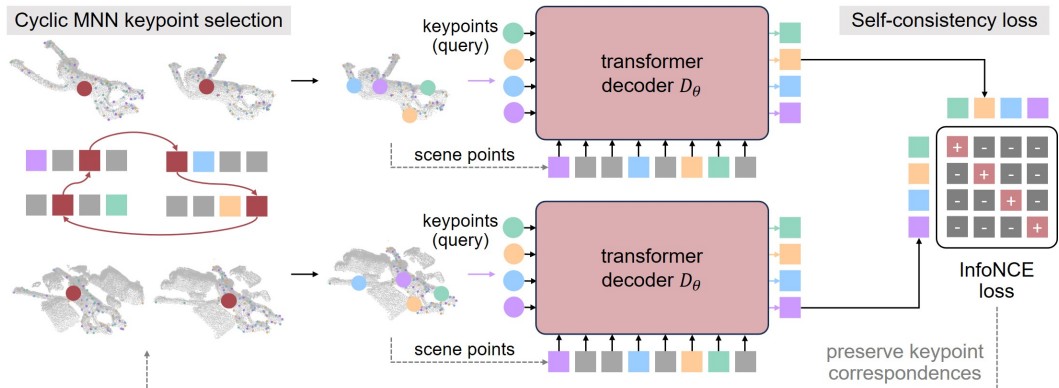

Figure 2: **Self-supervised training for the transformer decoder.** *Left:* Given a few scenes, we first select the corresponding keypoints computing cyclic mutual nearest neighbors (MNN) based on their feature similarities. *Right:* We take the selected keypoints as queries for each scene and enforce the features before and after the $D_\theta$-aggregation to induce the same keypoint correspondences across scenes. Specifically, we apply an InfoNCE loss to preserve the orders of the keypoints given the permutation equivariance of transformers.

While this spatial interpolation propagates the features from the discrete surface points to the near-surface space smoothly, this operation lacks semantic awareness and blurs the inter-scene feature correspondences as the query point goes off the object surfaces, making it unstable to transfer end-effector poses from the source scene to the target. For example, when transferring a grasp from a source scene containing only a toy animal to a target scene in which the toy animal is surrounded by several irrelevant objects, we aim to ensure that the features focus primarily on the toy animal itself and are less affected by the other objects despite their spatial adjacency (Fig. 1 top).

## 3.2 Neural Attention Field

To better represent dense and semantic-aware features in the near-object space, we introduce the *neural attention field* (Fig. 1) that predicts a cross-attention between the query point $\mathbf{q}$ and the scene points $\mathbf{X}$ to replace the inverse distance weighting $w_i$ in Eq. 2. More concretely, it is realized with a lightweight transformer decoder network $D_\theta(\texttt{key}, \texttt{query}, \texttt{value})$ with its keys, query, and value being the scene points $\mathbf{X}$, the query point $\mathbf{q}$, and the scene-point features $\mathbf{F}$ respectively:

$$\mathbf{f}_D = D_\theta(\mathbf{X}, \mathbf{q}, \mathbf{F}), \tag{3}$$

which provides an aggregated feature $\mathbf{f}_D \in \mathbb{R}^C$ for any query point $\mathbf{q} \in \mathbb{R}^3$ according to the learned attention maps. As the 3D point coordinates $\mathbf{X}$ and $\mathbf{q}$ carry limited information, we also append the per-point features $\mathbf{F}$ to $\mathbf{X}$ and the spatially aggregated feature $\mathbf{f}$ in Eq. 2 to $\mathbf{q}$, resulting in keys $[\mathbf{X}, \mathbf{F}]$ and query $[\mathbf{q}, \mathbf{f}]$. Detailed architecture of the transformer decoder can be found in the appendix.

## 3.3 Few-Shot Self-Supervised Training

As there is no straightforward way to annotate ground-truth features or attention maps, we propose a self-supervised framework for training the transformer decoder $D_\theta$ in Eq. 3 on a few 3D scenes related to the task. In most of our experiments, we use 4 scene pointclouds to train $D_\theta$ and then apply it to a set of tasks on transferring dexterous hand poses. We also experiment with different training and inference setups including different numbers of training scenes and different content similarities between the training and inference scenes (Sec. 4.2).

We consider directly using the scene points $\mathbf{x}_i \in \mathbf{X}$ as queries and compute their $D_\theta$-aggregated features via a self-attention:

$$\mathbf{f}_{D,i} = D_\theta(\mathbf{X}, \mathbf{x}_i, \mathbf{F}). \tag{4}$$

We denote $\mathbf{F}_D = \{\mathbf{f}_{D,i}\}_i$ the collections of aggregated features for all points in $\mathbf{X}$. Conceptually, this can also be viewed as placing a virtual query at each point in the scene pointclouds.

Given a few different scene pointclouds $\mathbf{X}^{(1)}, \cdots, \mathbf{X}^{(I)}$ with raw point features $\mathbf{F}^{(1)}, \cdots, \mathbf{F}^{(I)}$, we can compute their aggregated point features $\mathbf{F}_D^{(1)}, \cdots, \mathbf{F}_D^{(I)}$ respectively. Both $\mathbf{F}$ and $\mathbf{F}_D$ induce point correspondences across scenes via nearest neighbor retrieval *w.r.t.* feature similarities. Based on this, we introduce a self-supervised loss for training $D_\theta$ by enforcing that the induced correspondences from $\mathbf{F}$ and $\mathbf{F}_D$ should be identical between any pair of scenes (Fig. 2).

More concretely, we first compute a set of keypoints $\mathbf{K}^{(1)}, \cdots, \mathbf{K}^{(I)} \in \mathbb{R}^{K \times 3}$ for each scene based on the raw point features $\mathbf{F}^{(i)}$, with each keypoint indicating a correspondence across all scenes (*e.g.* the point on the monkey's left hand in all scenes). We then update the features of these keypoints via Eq. 4, acquiring their $D_\theta$-aggregated features $\mathbf{F}_D^{(i)}$. Finally, as transformer networks are permutation equivariant, we enforce the keypoint orders to stay consistent before and after applying $D_\theta$ with a contrastive loss on $\mathbf{F}_D^{(i)}$.

**Cyclic MNN keypoint selection (Fig. 2 left)** We compute the keypoints $\mathbf{K}^{(1)}, \cdots, \mathbf{K}^{(I)}$ such that each $\mathbf{K}^{(i)}$ is a subset of the points in $\mathbf{X}^{(i)}$ and any pair of $\mathbf{K}^{(i)}, \mathbf{K}^{(j)}$ are corresponding points across the two scenes according to the similarity of their raw features $\mathbf{F}^{(i)}, \mathbf{F}^{(j)}$. We introduce a cyclic mutual nearest neighbor (MNN) strategy for this corresponding keypoints selection, as illustrated in Fig. 2 left. Specifically, we randomly assign an order $1, \cdots, I$ to the scenes. For each adjacent scene pair $\mathbf{X}^{(i)}$ and $\mathbf{X}^{(i+1)}$ (or $\mathbf{X}^{(I)}$ and $\mathbf{X}^{(0)}$), any keypoint in $\mathbf{K}^{(i)}$ must fall into the k-nearest neighbor of the corresponding keypoint in $\mathbf{K}^{(i+1)}$ according to their feature differences. We reject any point that violates this constraint for any $i$.

**Self-consistency feature loss (Fig. 2 right)** The keypoint features are updated with $D_\theta$ via:

$$\mathbf{F}_D^{(i)} = D_\theta(\mathbf{X}^{(i)}, \mathbf{K}^{(i)}, \mathbf{F}^{(i)}), \;\; i = 1, \cdots, I. \tag{5}$$

We then enforce that, after this $D_\theta$ update, the feature-induced keypoint correspondences stay unchanged. Specifically, we employ an InfoNCE [48] loss as illustrated in Fig. 2 right:

$$\mathcal{L} = \sum_{i,j \in [I]} \mathcal{L}_{ij}, \quad \mathcal{L}_{ij} = \sum_{k=1}^{K} -\log \frac{\exp(\text{sim}((\mathbf{f}_{D,k}^{(i)}, \mathbf{f}_{D,k}^{(j)})/\tau))}{\sum_{k' \neq k} \exp(\text{sim}(\mathbf{f}_{D,k}^{(i)}, \mathbf{f}_{D,k'}^{(j)})/\tau)}, \tag{6}$$

where $\mathbf{f}_{D,k}^{(i)}$ is the $k$-th entry of $\mathbf{F}_D^{(i)}$, namely the output feature of $D_\theta$ of the $k$-th keypoint in scene $i$, and $\tau$ is a temperature parameter. This loss minimizes the updated feature differences of matched keypoints and maximizes that of unmatched keypoints. We optimize for the network parameters $\theta$ in $D_\theta$ *w.r.t.* this loss.

### 3.4 End-Effector Optimization

Following [6], we consider the dexterous hand parameters $\beta$ of the hand pose and joint rotations. Given a demonstration of hand parameters $\hat{\beta}$ in the source scene $\hat{\mathbf{X}}$ (with $\hat{\mathbf{F}}$), we optimize for $\beta$ in the target scene $\mathbf{X}$ (with $\mathbf{F}$) using the neural attention field. We start by sampling $Q$ points on the hand surfaces in both the source and target hands, generating query point sets $\mathbf{Q}|\hat{\beta}, \mathbf{Q}|\beta \in \mathbb{R}^{Q \times 3}$ conditioned on the hand parameters. The sampling is done in the canonical hand pose and is identical between the source and target hand surfaces. We follow the sampling strategy in [6] which has a higher sampling density on the fingers than on the palm.

We then obtain the features for the end-effector query points by passing them into the trained transformer decoder $D_\theta$ and acquire their features $D_\theta(\mathbf{Q}|\hat{\beta}, \hat{\mathbf{X}}, \hat{\mathbf{F}}), D_\theta(\mathbf{Q}|\beta, \mathbf{X}, \mathbf{F}) \in \mathbb{R}^{Q \times C}$. With these features, we optimize for the end-effector pose $\beta$ in the target scene while keeping all network parameters in $D_\theta$ frozen. The objective is to minimize the feature differences between the demonstration and target hand poses via an $l1$ loss, formulated as an energy function $E(\beta)$ *w.r.t.* $\beta$:

$$E(\beta) = \|D_\theta(\mathbf{Q}|\hat{\beta}, \hat{\mathbf{X}}, \hat{\mathbf{F}}) - D_\theta(\mathbf{Q}|\beta, \mathbf{X}, \mathbf{F})\|_1. \tag{7}$$

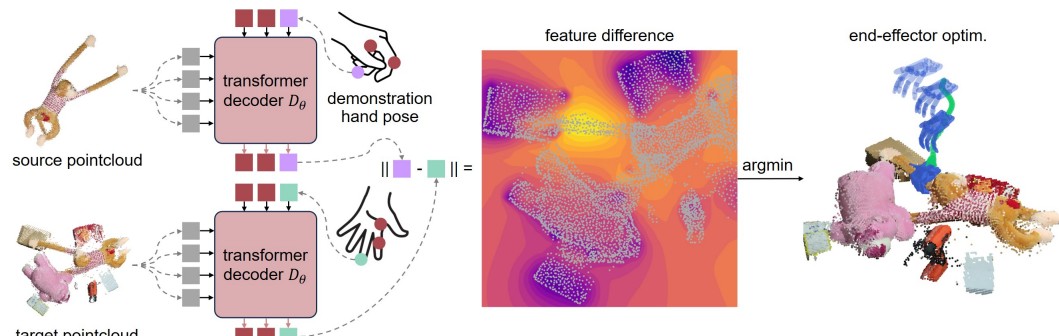

Figure 3: **End-effector optimization in the neural attention field.** *Left:* We sample query points on both the demonstration hand and the target hand to be optimized and obtain their features through the transformer decoder $D_\theta$. *Middle:* The feature differences induce an energy field. Yellow indicates lower energy values and thus higher feature similarities. *Right:* Minimizing the energy function *w.r.t.* the hand parameters gives the final grasping pose. Both hand positions and joint parameters are optimized. The optimization trajectory is shown in green and a few hand poses sampled along the trajectory are shown in blue (optimization steps indicated with colors from shallow to dark).

We also integrate the physical viability functions from [49, 6], including the inter-penetration and self-penetration energy functions:

$$E_{\text{pen}}(\beta|\mathbf{X}) = \sum_{\mathbf{x}\in\mathbf{X}} \mathbf{1}_{[\mathbf{x}\in\overline{Q}]}\text{dist}(\mathbf{x}, \partial\overline{\mathbf{Q}}), \quad E_{\text{spen}}(\beta) = \sum_{\mathbf{p},\mathbf{q}\in\mathbf{Q}|\beta} \mathbf{1}_{\mathbf{p}\neq\mathbf{q}}\max(\delta - \|\mathbf{p} - \mathbf{q}\|_2, 0), \quad (8)$$

where $\delta$ is a threshold parameter, and a pose constraint $E_{\text{pose}}(\beta)$ that penalizes out-of-limit hand pose as in [49]. The overall optimization combines these terms:

$$E(\beta) + \lambda_{\text{pen}}E_{\text{pen}}(\beta|\mathbf{X}) + \lambda_{\text{spen}}E_{\text{spen}}(\beta) + \lambda_{\text{pose}}E_{\text{pose}}(\beta). \quad (9)$$

The weighting between feature energy loss and the physical constraints are set to be $\lambda_{\text{pen}} = 10^{-1}, \lambda_{\text{spen}} = 10^{-2}, \lambda_{\text{pose}} = 10^{-2}$, which is the same as [6]. In all our experiments, we randomly initialize the hand position and joint parameters and optimize for via gradient descent.

## 4 Experiments

**Training and inference** In all our experiments, we pre-train the transformer decoder network $D_\theta$ with a few scene pointclouds (2-4 scenes without demonstrations). We then provide a 1-shot demonstration with a dexterous hand pose in a 3D scene and directly transfer it to different scenes. This demonstration scene pointcloud doesn't have to be included in the pre-training scenes. We experiment with different pre-training and demonstration scene setups as listed in the tables. Each evaluation is done 10 times with scene variations in the object poses (for both single-object and multi-object scenes).

**Real-robot setup** We conduct all our experiments in the real world with a Shadow Dexterous Hand of 24 DoF. We restrict the 2 DoF at the wrist, focusing the optimization on the remaining 22 DoF. The hand is mounted on a UR10e arm, introducing 6 additional DoF. Precautions are taken to prevent the dexterous hand from contacting the table surface directly, and the range of motion is limited by the arm's elbow. For RGBD scans, we use four pre-calibrated Femto Bolt sensors at each corner of the table. Post-processing is applied to the captured scans to remove background elements: In single-object experiments, the object's pointcloud is segmented using SAM [50]. For multi-object scenarios, we apply physical constraints to limit the experimental area to the tabletop and employ RANSAC [51] to exclude the table surface from the analysis.

**Baselines** We mainly compare our method to the feature-field-based methods, including a vanilla DFF with the feature acquisition strategy in [25], and SparseDFF [6] which introduces an additional multiview feature refinement mechanism for better feature qualities.

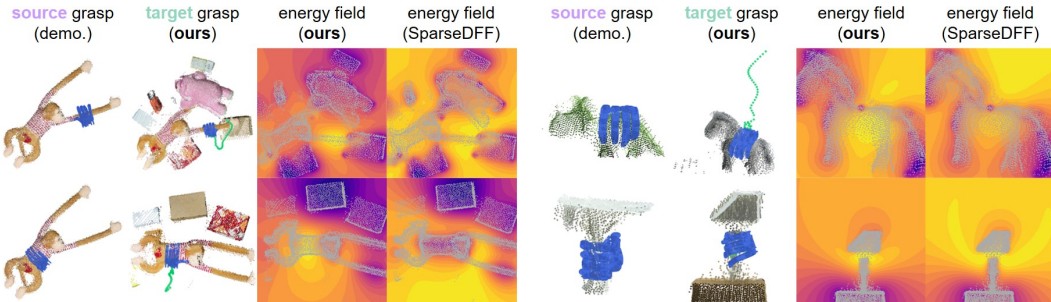

Figure 4: **Visualizations of the energy fields and end-effector optimization.** In each group from left to right are: the source scene with demonstration (hand shown in blue); the target scene and our results (optimization trajectory shown in green and final resulting hand shown blue); the feature-induced energy fields for our method and SparseDFF [6]. We only visualize the 2D sections of the 3D energy fields and yellow indicates lower energy values and thus higher feature similarities. Our method shows more concentrated low-energy regions around the target grasping positions.

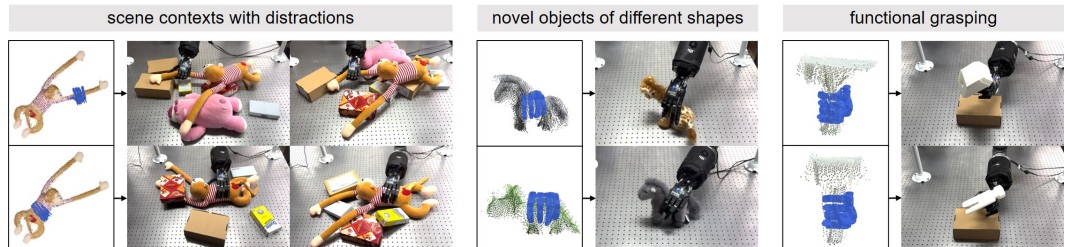

Figure 5: **Real-robot results.** From left to right, we show our results of grasping objects in scene contexts with distraction, grasp transfer between objects with similar semantics but shape variations, and functional grasping of tools.

## 4.1 Real-Robot Results

**Object in scene contexts**    A major advantage of our neural attention field is that the cross-attention mechanism between end-effector queries and the scene points encourages the focus on the scene points of higher relevance to the task indicated by the demonstration. This is particularly beneficial when there are task-irrelevant objects in the scene and we want to accurately locate specific regions on the target object and avoid distractions. Tab. 1 shows our results for grasping the target objects from multi-object scene contexts with only a one-shot grasping demonstration on a single object. Our method has consistently higher success rates than the previous feature-field-based methods. Real-robot executions are shown in Fig. 5 left.

Fig. 4 visualizes the energy fields of our method and a vanilla distilled feature field method SparseDFF [6]. We can see that our attention field induces energy functions with more concentrated minimas centered around the target grasping positions (such as the monkey arm in the top left example), while the distilled feature field induces more irregular energy functions in the space, resulting worse optimization landscapes.

**Cross-object grasp transfer**    Tab. 2 shows our results on dexterous grasp transfer between objects of varying shapes but similar semantics. In addition to filtering out the irrelevant content in multi-object scenes, our attention field also brings benefits even in single-object scenes by encouraging the transformer queries to focus on task-related object regions. For example, the necks of the horse and the giraffe are of very different shapes and are close to the back where the grasp is applied, but semantically they are irrelevant to the task. Simply computing the features according to spatial relations as in the feature fields can be misleading, while our attention field overcomes this issue. Real-robot executions are shown in Fig. 5 middle.

Table 1: Object in scene contexts.

| $D_\theta$ pre-train | Demo scene | Test scene | Ours | SparseDFF | DFF |
|---|---|---|---|---|---|
| 4 monkey | monkey right arm | monkey in context | **10/10** | 6/10 | 4/10 |
| 4 monkey | monkey back | monkey in context | **10/10** | 8/10 | 5/10 |
| 4 monkey | monkey back | horse in context | **9/10** | 3/10 | 0/10 |

Table 2: Cross-object grasp transfer.

| $D_\theta$ pre-train | Demo scene | Test scene | Ours | SparseDFF | DFF |
|---|---|---|---|---|---|
| 1 horse, 1 rhino | horse back | giraffe | **9/10** | 5/10 | 0/10 |
| 1 horse, 1 giraffe | giraffe back | rhino | **10/10** | 2/10 | 0/10 |
| 1 rhino, 1 giraffe | rhino back | horse | **10/10** | 7/10 | 0/10 |
| 1 rhino, 1 dinosaur | horse back | giraffe | **7/10** | 5/10 | 0/10 |
| 4 monkey | horse back | giraffe | **9/10** | 5/10 | 0/10 |

Table 3: Functional grasping.

| $D_\theta$ pre-train | Demo scene | Test scene | Ours | SparseDFF | DFF |
|---|---|---|---|---|---|
| 1 nail hmmr., 1 rubber mallet | nail hmmr. | Thor hmmr. | **9/10** | 8/10 | 8/10 |
| 1 nail hmmr., 1 Thor hmmr. | Thor hmmr. | rubber mallet | **8/10** | 4/10 | 0/10 |
| 1 Thor hmmr., 1 rubber mallet | rubber mallet | nail hmmr. | **9/10** | 5/10 | 0/10 |

**Functional grasping** The semantic awareness in our neural attention also facilitates functional grasping. Tab. 3 shows our results for functional grasping of tools where the demonstrations and executions are on tools with varying shapes but similar functionalities. Real-robot executions are shown in Fig. 5 right.

## 4.2 Ablation Studies

Now we conduct ablation studies for the $D_\theta$ pre-training setups on the number of pre-training scenes and whether the demonstration scene is included. All ablation experiments are done with the monkey toy. Fig. 6 plots the results of our method under different numbers of $D_\theta$ pre-training scenes. Our method shows consistent performances over these different pre-training setups. Qualitative results can be found in the appendix.

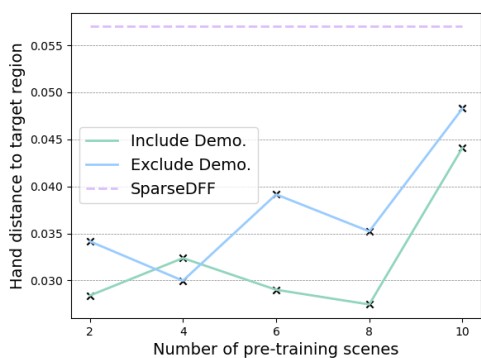

Figure 6: $D_\theta$ **pre-training with different setups.**

## 5 Conclusions

To better model hand-object interactions, we propose the neural attention field, which represents semantic features in the 3D space via modeling inter-point relevance instead of individual point features. It enables robust and semantic-aware transfer of dexterous grasps across scenes by encouraging the end-effector to focus on scene regions with higher task relevance.

**Limitations and future work** While we focus on dexterous grasps in this work, the neural attention field and its subsequent feature-based end-effector optimization can be applied to other dexterous manipulations beyond grasping. Extension to other manipulations with action sequences would be an interesting and useful future direction.

**Acknowledgments**

This work was supported in part by the Toyota Research Institute, the National Science Foundation under Grant Number 2342246, and by the Natural Sciences and Engineering Research Council of Canada (NSERC) under Award Number 526541680.

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

# A    Network Details

**Transformer decoder architecture**    Fig. 7 illustrates the network architecture of our transformer decoder. We compute the pairwise distances between the hand points and scene points and apply them as weights to the keys and queries for more stable convergence during training.

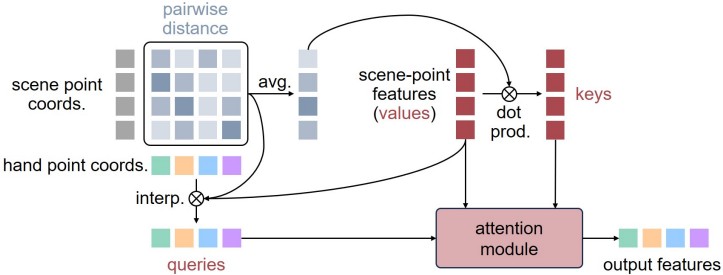

Figure 7: **Network architecture of the transformer decoder.**

**Network training**    The transformer decoder network is only trained at the feature pre-training stage with the self-consistency losses, and all network weights are frozen afterward in the end-effector optimization stage. All networks are trained on a single GeForce RTX 3090 with an Adam optimizer for 100 iterations.

# B    Real-Robot Setup Details

Fig. 8 shows our real-robot setup. Four Femto Bolt sensors are mounted at the four corners 100cm above the table, with each edge 100cm. The objects are placed near the center of the table. The dexterous hand is mounted on a UR10e arm, reaching the objects from the left.

**Shadow hand parameterization**    As mentioned in the paper, we conduct all our experiments in the real world with a Shadow Dexterous Hand of 24 DoF. We restrict the 2 DoF at the wrist, focusing the optimization on the remaining 22 DoF. The UR10e arm also introduces 6 additional DoFs. Each hand joint is parameterized as a scaler for its rotation angle. The

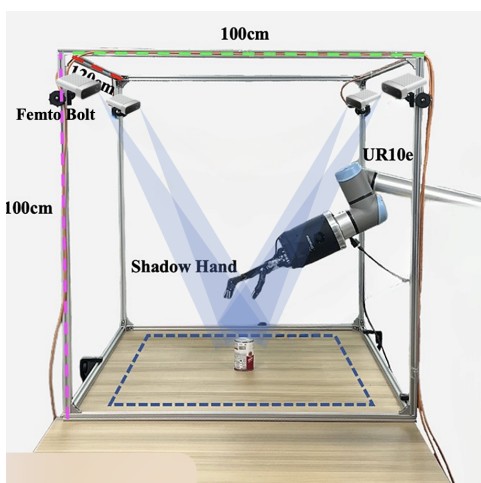

Figure 8: **Real-robot setup.**

shadow hand has 4 under-actuated joints. But in our method, we just regard it as a normal actuated joint, which is a common practice and the default API from the shadow hand takes these four joints as normal joints. We use "rotate6D" to represent the 3-DoF hand base rotation, which is the first two columns of the rotation matrix.

**Scene setup and randomization**    In the real-robot experiments, for single-object scenes with toy animals, we randomize its pose with arbitrary z-axis rotations so that the dexterous hand can grasp it from the top without hitting the table. For functional grasping with 3D-printed tools, we put them on boxes so that the robot can reach its handle without hitting the table. For multi-object scenes, we randomize the poses of all objects with full $SO(3)$ rotations.

# C    Ablation Study Details

**Evaluation metrics (Fig. 6)**    The ablation studies are done virtually on pre-captured 3D scene pointclouds to eliminate the variances caused by scene randomization, and thus the 10 evaluation

scenes in all the ablation settings are identical. In Fig. 6, we evaluate the average distance between the hand-surface query points and the target region (monkey arm) in the scene, as a rough metric indicating how close the hand reaches the target post-optimization. The target region is manually annotated with MeshLab.

**Quantitative results** Fig. 9 shows the qualitative results on grasping the monkey arm with the same demonstration but different pre-training setups for the transformer decoder. The hand optimization trajectories are shown in green.

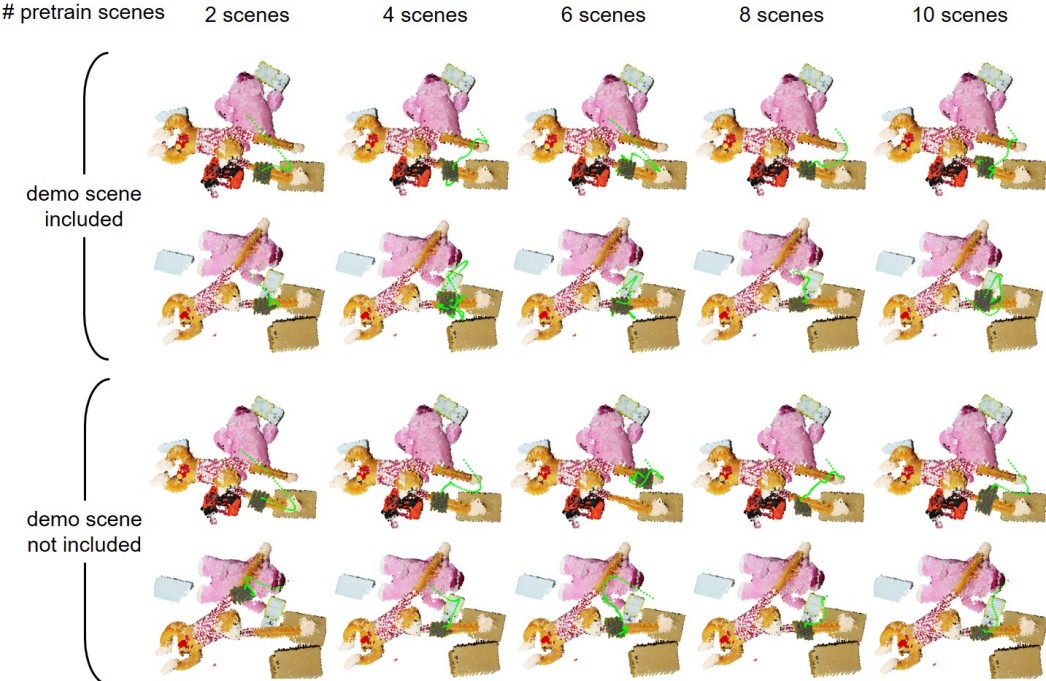

Figure 9: **Qualitative results for the ablation study on different pre-training setups.**

