# OpenReview forum: "Neural Attention Field: Emerging Point Relevance in 3D Scenes for One-Shot Dexterous Grasping"
_robot-learning.org/CoRL/2024/Conference — CoRL 2024_

### Official Review · Reviewer_vfCH · 2024-07-18
**Neural Attention Field: Emerging Point Relevance in 3D Scenes for One-Shot Dexterous Grasping**

**Originality:** 3
**Technical Quality:** 3
**Clarity Of Presentation:** 2
**Potential Impact:** 2
**Recommendation:** 3
**Confidence:** 4

**Review:**

Strengths
- Semantic-Aware Features: The neural attention field enables free space pose optimization for hand grasping by creating semantic-aware features. This allows the model to focus on task-relevant regions of the scene, improving the accuracy and robustness of grasping tasks.
- Self-Supervised Learning: The proposed self-supervised learning framework facilitates efficient learning from a few samples, reducing the need for extensive labeled datasets. This approach enhances the practicality and scalability of the method.
- Improved Performance: Experimental results demonstrate that the method significantly outperforms baseline feature-field-based methods. The neural attention field provides better optimization landscapes, leading to higher success rates in dexterous grasping tasks.
- Generalization: The method shows strong generalization capabilities, successfully transferring grasps to new objects and scenes with varying contexts. This robustness is critical for real-world applications where the environment can be highly dynamic and unpredictable.


Weaknesses
- Lack of Detail on Design Choices: The paper lacks detailed explanations on certain design choices and implementation aspects, such as the method for pairing key points across scenes (lines 151-152) and the definition of Q in equation 8 .
- Potentially Costly Optimization Runtime: The optimization process, especially with the transformer-based approach, could be computationally expensive, leading to longer runtimes and potentially limiting its practicality for real-time applications.
- Expensive Data Labeling Efforts: Despite the self-supervised learning framework, significant data labeling efforts may still be required, particularly for soft or similar objects, which could limit scalability.
- Clarity and Readability: The paper could improve in clarity, especially in explaining the calculation of scene features (Appendix Figure A) and the impact of query point initialization. Enhancing these explanations would aid reader understanding.
- Performance in Occluded Scenes: The paper does not thoroughly evaluate the method's performance in scenes with occlusions, which could significantly impact the effectiveness of the proposed approach. Further experiments are needed to assess robustness in such scenarios.

**Quality Of The Limitations Section:**

3

**Questions For Rebuttal:**

Questions for Rebuttal
- How does this method transfer to scenes with unseen objects as distractors?
- How is the pairing of key points across scenes achieved (lines 151-152)? Does this method require a curated dataset with cross-scene point pairs?
- What is $\bar{\mathbb{Q}}$ in equation 8 $E_{pen}$?
- Does the initialization of query points (Q) affect the quality of the final grasp pose?
- How are the scene features calculated, as shown in Appendix Figure A?
- How does the method perform in scenes with occlusions?

**Robotics Focus:**

4

**Summary Of Paper:**

The paper introduces a neural attention field based on an attention decoder to create semantic-aware features for one-shot dexterous grasping transfer. This approach models inter-point relevance instead of individual point features, enhancing the ability to generalize grasps across different 3D scenes. The authors propose a self-supervised framework for training the attention decoder, eliminating the need for extensive hand-labeled datasets. Experimental results demonstrate that this method significantly improves performance over baseline feature-field-based methods, showing better optimization landscapes and higher success rates in grasping tasks with real robots.

**Summary Of Recommendation:**

The paper introduces a neural attention field for one-shot dexterous grasping, using a transformer decoder to enhance semantic understanding. The self-supervised learning approach enables efficient training from few samples and shows significant performance improvements over baselines. However, the paper lacks detail on key design choices and does discuss potential costly runtime and data collection process. Despite these issues, the contributions and results justify acceptance.

---

### Official Review · Reviewer_36Qt · 2024-07-20
**Concerns about the Real-World Experiments and differences to SparseDFF**

**Originality:** 4
**Technical Quality:** 4
**Clarity Of Presentation:** 5
**Potential Impact:** 3
**Recommendation:** 2
**Confidence:** 4

**Review:**

The paper is very well written, and the figures nicely illustrate the method.

My major concern with this work is that it is very similar to [6]. It would be good if the authors could highlight the differences more. The authors write in the introduction that SparseDFF “is prone to failure when there are distractors.” Couldn’t this be addressed by adding a segmentation step method and removing the distractors in the scene? It would be good to see some experiments on this.

Other than that, it seems like the objects in the real-world experiment “Monkey to horse context” are placed much more apart than for the baselines. This does not really support the claim that SparseDFF “is prone to failure when there are distractors” in the scene. I would like to see some plots about the spatial distribution of the objects. SparseDDF also shows some experiments when grasping the Monkey in clutter and reports 100% success rate. Why the drop if grasping is restricted to the arm? Furthermore, when grasping the mallet, it seems that in the experiments it was placed in a similar manner but rotated for the baselines. Why does SparseDFF perform worse here if there are not that many distractors?

## Minor:

-	Line 18: Desterous Grasping
-	Line 162: missing )
-	Equation 6: additional (
-	The rotation animation in the video at 1:58 is unnecessary and confusing
-	It would be good to have labels for the energy field in Fig. 3 and Fig. 4 to help the reader to understand what the colors mean. The video narration mentions this, but it would be good to have a label.

**Quality Of The Limitations Section:**

2

**Questions For Rebuttal:**

1.) Why not simply add a segmentation step to [6]? This would filter out the distractors.

2.) The clutter in the real-world experiment seems to be biased towards this method. It seems like objects are placed more apart in the “Monkey to horse context.” E.g., the power drill in the video at 2:45 is. Can you comment on that?

3.) Actually, can you redo the “Monkey to horse context” experiment and place the objects close together as you did for SparseDFF?

4.) Similarly, the rubber mallet in 2:45 seems to be rotated less than for the baselines. Why does SparseDFF only perform 50% if there are few distractors?

5.) Can you elaborate more on the task-relevance of this feature?

**Robotics Focus:**

3

**Summary Of Paper:**

This paper presents a neural attention field method for one-shot transfer of dexterous grasps. The work builds on previous work [6] by shifting the focus to task-relevant scene points instead of spatial proximity. The work is evaluated in real-world scenarios using a shadow hand.

**Summary Of Recommendation:**

Dexterous grasping is challenging, but I have some concerns about the experiment setup, as objects are placed much further apart, questioning the claim that this method is more robust to distractors. I am looking forward to the rebuttal.

---

### Official Review · Reviewer_gKTk · 2024-07-25
**Significant Improvement Needed in Experimental Quality and Contributions**

**Originality:** 2
**Technical Quality:** 2
**Clarity Of Presentation:** 4
**Potential Impact:** 2
**Recommendation:** 1
**Confidence:** 3

**Review:**

Strength:
- Success utilization of a transformer to model inter-point relevance.
- Successfully demonstrated a self-supervised training framework for the transformer decoder that requires only a few 3D scenes and reducing dependency on large annotated datasets.
- Demonstrates real-robot grasping with one-shot demonstration.

Weakness:
- The paper’s contribution is relatively limited, primarily adding an extra layer to an existing method. The reviewer expected more comprehensive application results beyond simple grasping experiments on just a few objects.
- The lack of state-of-the-art baselines for comparison makes the current experiment resemble an ablation test, merely showing the improvement of adding the neural attention field on top of DFF. It fails to compare with other widely used object-centric or semantic-aware frameworks for visual manipulation.
- The training and evaluation scope is too narrow to draw meaningful conclusions about the effectiveness and generalizability of the proposed method.

**Quality Of The Limitations Section:**

1

**Questions For Rebuttal:**

From an applied robotics perspective, I am curious how the authors envision this method affecting downstream applications. Specifically, in a problem setting where the model is trained on grasping one or a few objects and then applied to grasp the same or similar objects in more complex environments. Although fast adaptation is beneficial, the applicability seems limited because new objects with new semantics would still require new demonstrations for re-training.

**Robotics Focus:**

2

**Summary Of Paper:**

This work introduces a neural attention field using a transformer decoder to model inter-point relevance, enhancing feature representation and focusing on task-relevant regions. They uses a self-supervised framework to train the model on a few 3D scenes and applied to new scenes for grasping purposes from one-shot demonstrations.

**Summary Of Recommendation:**

Recommend a rejection unless followup draft demonstrates a significant improvement in experimental quality and contributions

---

### Author Rebuttal · Authors · 2024-08-14

**New Experiments**
We appreciate the insightful feedback from the reviewers. In this rebuttal comment, we attach a PDF titled ("rebuttal.pdf", attached as .zip file) with additional experiments and visualizations that are discussed in our response to Reviewer 36Qt.

---

### Decision · Program_Chairs · 2024-09-04

**Decision:**

Accept

**Comment:**

The paper introduces a new method for one-shot grasp transfer for dexterous grasping. A key component is the introduced neural attention field to capture dependencies between 3D points by using an attention mechanism. Grasps are optimized using features from the  neural attention field.

Strengths

- The reviewers agree on that the neural attention field for one-shot grasping is new.
- The paper is well written and easy to read.

Weaknesses

The reviewers have raised several concerns about the paper.

- Limited novelty of the paper regarding the difference between the proposed method and [6].
- Limited experimental evaluations. No comparison with other state-of-the-art baselines. Can object segmentation be used with SparseDFF?
- Clarification of certain experimental setup.

Post-Rebuttal
- The authors provided their answers to address the concerns from the reviewers. However, the reviewers did not participant in the discussion with several messages from the AC.
- The AC checked the rebuttal and agrees with the authors. The proposed method has enough novelty to address the one-shot grasping problem for dexterous hands. It shows improvement over SparseDFF [6].